# OODREB: Benchmarking State-of-the-Art methods for Out-Of-Distribution Generalization on Relation Extraction

Anonymous

## ABSTRACT

Relation extraction (RE) methods have achieved striking performance when training and test data are independently and identically distributed (*i.i.d* assumption). However, in real-world scenarios where RE models are trained to acquire knowledge in the wild, the assumption can hardly be satisfied due to the different and unknown testing distributions. In this paper, we serve as the first effort to study out-of-distribution (OOD) problems in RE by constructing an out-of-distribution relation extraction benchmark (OODREB) and then investigating the abilities of state-of-the-art (SOTA) RE methods on OODREB in both *i.i.d.* and OOD settings. Our proposed benchmark and analysis reveal new findings and insights: (1) Existing SOTA RE methods struggle to achieve satisfying performance on OODREB in both *i.i.d.* and OOD settings due to the complex training data and biased model selection method. Rethinking the developing protocols of RE methods is of great urgency. (2) The SOTA RE methods fail to learn causality due to the diverse linguistic expressions of causal information. The failure limits their robustness and generalization ability; (3) Current RE methods based on language models are far away from being deployed in real-world applications. We appeal to future work to take the OOD generalization and causality learning ability into consideration.

## KEYWORDS

Relation extraction, out-of-distribution generalization, benchmark

**ACM Reference Format:**
Anonymous. 2018. OODREB: Benchmarking State-of-the-Art methods for Out-Of-Distribution Generalization on Relation Extraction. In *Proceedings of Make sure to enter the correct conference title from your rights confirmation emai (Conference acronym 'XX).* ACM, New York, NY, USA, 11 pages. https://doi.org/XXXXXXX.XXXXXXX

## 1 INTRODUCTION

Relation extraction (RE), aiming to extract relational facts from the given context, facilitates a wide range of downstream tasks and applications, including text summarization [12, 23], question answering [11], and natural language inference [1, 31]. Most prevalent RE methods are based on supervised learning. They typically minimize their training errors by greedily absorbing all correlations discovered in data, which leads to a series of issues. The most

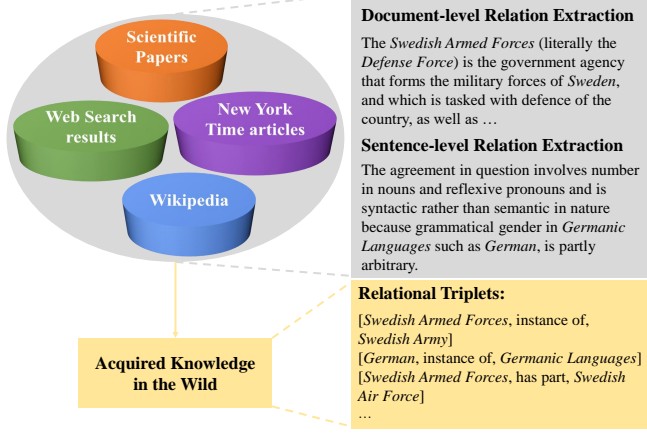

**Figure 1: A brief illustration of knowledge acquisition in the wild. The unstructured text data from different domains and sources exhibit various complex distributions, violating the underlying *i.i.d.* assumption of current RE methods.**

frequently reported one is that RE models can often spuriously correlate entity names with the final prediction [6, 24, 32]. Although their learned spurious correlations can be predictive in the test set due to the *i.i.d.* assumption which assumes that the training and test data are independently and identically distributed, most of the spurious correlations will no longer hold in the out-of-distribution (OOD) setting, where RE methods are exposed to real-world data with different distributions. Since the ultimate goal of the RE task is to acquire knowledge in the wild [34], tackling the understudied OOD generalization issue in RE is of great urgency.

The main challenge that impedes previous work from studying the OOD generalization (knowledge acquisition in the wild) ability of RE methods is the lack of a benchmark. Previous work [10, 34, 35] constructs benchmarks in RE in several typical stages: (1) Researchers first generate distantly supervised annotation for their collected data; (2) human annotators perform annotating on the generated annotation according to the prescribed entity and relation types. The construction is not centered on collecting various distributions of data for each relation type.

Due to the long-term neglect and lacking consideration of the OOD generalization problem in RE, few current RE methods are possible to acquire knowledge in the wild. As shown in Figure 1, real-world textual data comprise both document- and sentence-level samples from various domains with different writing styles. Only the document-level relation extraction (DocRE) methods [33, 35, 41] and the recently proposed LLMs [22] are expected to be able to tackle the challenging issue: They are capable of performing both intra- and inter-sentence reasoning and processing long inputs. Therefore, we urgently require a benchmark to investigate the

current models, present the gap between their striking task performance and the understanding of their OOD generalization behavior, and provide detailed analysis to facilitate future development on real-world RE.

In this paper, we serve as the first effort to study the OOD generalization ability of RE methods. We conduct human annotation to propose an out-of-distribution relation extraction benchmark (OODREB) to simulate real-world scenarios. Based on OODREB, we rethink the whole developing protocol of real-world RE and shed some light on the following crucial yet rarely raised questions. **Q1**: Can we easily solve real-world (OOD) RE problems by involving as many training data from various distributions as possible? We discuss this in Section 6.4. The answer is probably no due to both the complexity of data that increases the difficulty in learning causality and the limitations of model architectures and model selection methods, which suggests the rethinking of developing protocols in RE. **Q2**: Can we apply existing SOTA RE methods to solve real-world RE problems? We investigate the problem in Section 6.5 and show the severe lack of robustness and generalization ability of existing methods. **Q3**: Given that models perform well on *i.i.d.* data yet exhibit poor performance on OOD data, what are the underlying causes of the performance decline during the transition from *i.i.d.* to OOD data? We elaborate on the details in Section 6.7 and show the phenomenon that the larger the shift, the poorer the performance. The underlying cause is their failure to learn causality. **Q4**: Why do models fail to learn causality? We investigate model performance across all relation types in Section 6.8 and then discover a common feature of the challenging relation types: They possess diverse linguistic expressions of their causal semantic features. To sum up, we appeal to future work to rethink the development protocols of RE methods and take the OOD generalization ability of models into consideration.

**Observations.** Our findings are summarized as follows: (1) Expanding human-annotated training data from various distributions does not improve the ability of RE methods to acquire knowledge in the wild due to the limitation of model architectures and model selection methods. (2) Existing SOTA DocRE methods fail to capture causal information and thus present weak robustness and poor generalization ability on OODREB. (3) If a model fails to learn causality, a larger distributional shift leads to its sharper performance drop. (4) The diverse linguistic expressions of causal information render it more challenging to learn causality. (5) When the experimental situation transits closer to the most common real-world scenario faced by RE practitioners, the OOD generalization ability of models gets poorer. (6) We find clues that the representative ChatGPT [22] struggles to both make accurate predictions (0.921% F1 score on the validation set of DocRED) and conform to the RE requirements.

## 2 RELATED WORK

**Out-Of-Distribution Generalization.** OOD generalization is proposed to improve the OOD generalization ability of models under distribution shifts [7, 18, 28]. Existing methods tackle OOD generalization problems in three ways [29]: (1) They try to formally characterize the distribution shifts. Specifically, domain generalization methods model the distribution shifts as the data collected from

different domains [17], while casual or stable learning methods employ causal structure models to reveal the inherent mechanisms that cause distribution shifts [8]. (2) There is a surge of attempts to propose OOD algorithms by either unsupervised representation learning [15], supervised learning [17], or optimization [18]. (3) They propose specific datasets and the corresponding evaluation metrics to evaluate the OOD performance of various methods, including perturbing the data generation process [16] and introducing real-world data like CivilComments [21] and Amazon [3]. To the best of our knowledge, this is the first work to tackle the OOD problems in the RE task comprehensively.

**Relation Extraction.** Relation Extraction aims to extract relational facts from the given context, which is a crucial step toward automatic KB construction [26]. It provides the extracted knowledge as a supplement to many studies, including text summarization [12, 23], question answering [11], and natural language inference [1, 31]. Existing RE tasks can roughly be divided into two categories according to the input context: sentence- and document-level RE. The major challenge in sentence-level RE is the overlapping relational facts [37] while in document-level RE is intra- and inter-sentence reasoning [35]. Since a document comprises multiple sentences, document-level RE (DocRE) methods are required not only to conquer the challenges in sentence-level RE but also to satisfy the need of document-level RE. Therefore, we mainly investigate the performance of DocRE methods on OODREB. To the best of our knowledge, this is the first work to study OOD generalization problems in RE to facilitate real-world RE.

## 3 PROBLEM FORMULATION

Given a text $\mathbf{x} = \{x_1, x_2, \ldots, x_n\}$ that comprises $n$ sentences with $s$ entities $\mathbf{e} = \{e_1, e_2, \ldots, e_s\}$. The semantic space and the label space are $\mathcal{X} \in \mathbb{R}^h$ and $\mathcal{Y} \in \{y_1, y_2, \ldots, y_t\}$, respectively. We aim to find a labeling function $f : \mathcal{X} \to \mathcal{Y}$ to accurately map the given text $\mathbf{x}$ to its labels $\mathbf{y}$ according to the current entity pair $(e_i, e_j)_{i,j=1,\ldots,s;i\neq j}$. There can be multiple relations (up to the prescribed $t$ relation types) between an entity pair.

To further describe the core of real-world RE, we define the decision rule. We use set $\mathcal{D}\left(f, \mathbf{x}, (e_i, e_j)\right)$ to describe the features used by function $f$ when given the text and an entity pair. Note that there are multiple candidate formulas to define $\mathcal{D}$, which depends on further mathematical discussions. For example, previous work [4, 5, 25, 36] in domain generalization often opt to narrow the concept of $\mathcal{D}$ and defines it as a minimum sufficient explanation to quantify the error bound. The decision rule of human can be described by $\mathcal{D}\left(f_h, \mathbf{x}, (e_i, e_j)\right)$ and the decision rule of model is $\mathcal{D}\left(f_m, \mathbf{x}, (e_i, e_j)\right)$. For any $\mathbf{x}$, $\mathbf{y} := f_h\left(\mathbf{x}, (e_i, e_j)\right)$. During the training process where the training data is sampled from the distribution $\mathbf{P}_a$, we may obtain $f_m$ that for any $\mathbf{x} \sim \mathbf{P}_a$, $f_h(\mathbf{x}) = f_m(\mathbf{x})$. In *i.i.d.* setting, the RE task is considered solved if we find such a $f_m$ regardless of whether $f_m$ and $f_h$ share the common decision rule $\mathcal{D}$.

**Causality, Stable patterns, and Spurious Patterns.** In real-world scenarios, $\mathbf{x}$ can come from other distributions (e.g., $\mathbf{P}_b$). We consider $f_m$ as learning spurious patterns if $\mathcal{D}\left(f_h, \mathbf{x}, (e_i, e_j)\right)$ and $\mathcal{D}\left(f_m, \mathbf{x}, (e_i, e_j)\right)$ are different. Since $\mathbf{y} := f_h\left(\mathbf{x}, (e_i, e_j)\right)$, the performance of $f_h$ is stably accurate on any data across all distributions due to its used features $\mathcal{D}\left(f_h, \mathbf{x}, (e_i, e_j)\right)$. Our goal is to

| Statistics | CoNLL04 | FewRel | KBP-37 | SciERC | SemEval | TACRED | DocRED |
|---|---|---|---|---|---|---|---|
| Samples | 980 | 35,000 | 15,768 | 409 | 2,227 | 11,802 | 4,051 |
| Relation types | 3 | 50 | 9 | 2 | 2 | 18 | 96 |
| Relational facts | 1,372 | 35,000 | 15,768 | 675 | 2,227 | 11,802 | 50,455 |
| Avg. relational facts | 1.40 | 1.0 | 1.0 | 1.66 | 1.0 | 1.0 | 12.45 |
| Avg. words | 29.40 | 25.28 | 44.62 | 29.06 | 20.22 | 34.57 | 198.35 |
| Avg. entities | 3.64 | 2.0 | 2.0 | 4.58 | 2.0 | 2.0 | 19.50 |
| Avg. sentences | 1.0 | 1.0 | 1.0 | 1.0 | 1.0 | 1.0 | 7.97 |

**Table 1: Statistics of OODREB. "SemEval" denotes the sentence-level RE dataset SemEval 2010 Task 8. "Avg." stands for the average number of the corresponding items per sample.**

find a function $f_m$ that is as accurate and stable as $f_h$. In other words, learning stable patterns is an effective way to avoid learning spurious patterns and approach learning causality (for any **x** from any distributions, $f_h(x) = f_m(x)$). In the RE task, a structural causal model proposed by recent work [6] reveals that both the pre-training and finetuning processes can hamper models from learning stable patterns.

**The function of OODREB.** Our work provides the first benchmark, which can also be construed as an exam, to evaluate whether a RE model can be deployed in real-world scenarios. If a RE model fails to pass our exam, it definitely cannot work well in a wide range of real-world scenarios. One might ask what is the most ideal model that can directly be deployed in real-world scenarios? We give the definition: $f_h(x) = f_m(x)$. We hope that there exists an AI model, trained by any means, is able to pass our exam, enlighten RE practitioners, and facilitate real-world scenarios instead of merely chasing the SOTA performance in the *i.i.d.* setting without considering $f_h(x)$ and real-world scenarios. Recent work [6] experimentally reveals that $f_h(x)$ should be word-level evidence in DocRE and provides a metric to measure the distance between $f_h(x)$ and $f_m(x)$.

## 4 DATASETS

Table 1 shows the statistics of our constructed OODREB, which consists of samples from 7 human-annotated RE datasets: DocRED [35], CoNLL04 [27], FewRel [13], KBP-37 [38], SciERC [20], SemEval 2010 Task 8 [14], and TACRED [40]. Details of the 7 datasets are as follows.

**DocRED.** DocRED [35] is proposed for document-level RE, consisting of 3,053 human-annotated training instances, 1,000 development instances, and 1,000 testing instances. Over 61.1% relations in DocRED require reasoning. The annotation of the test set is not publicly available yet. Therefore, we select samples from the training set and validation set of DocRED. All documents are derived from Wikipedia.

**CoNLL04.** CoNLL04 [27] is proposed for sentence-level RE and consists of sentences from news articles with five relation types and comprises 1,290 samples in the training set, 343 samples in the validation set, and 422 samples in the test set.

**FewRel.** FewRel [13] is composed of 70,000 sentences on 100 relations and is proposed for sentence-level few-shot RE. All of the sentences are derived from Wikipedia.

**KBP-37.** KBP-37 [38] is a revision of the MIML-RE annotation dataset, containing 15,917 samples in the training set, 1,724 samples in the validation set, and 3.405 samples in the test set with 37 relation types. The sentences are derived from The New York Times articles.

**SciERC.** SciERC [20] contains annotations for scientific entities and their relations for 500 scientific abstracts. The abstracts are selected from 12 AI conference/workshop proceedings. The dataset includes 4,716 relations with 7 relation types for sentence-level RE.

**SemEval 2010 Task 8.** SemEval 2010 Task 8 [14] is proposed for sentence-level RE and contains 10,717 instances with 9 relations. The samples are manually collected by pattern-based Web search.

**TACRED.** TACRED [40] comprises 119,474 examples in 41 relation types. The dataset is proposed for sentence-level RE, the examples of which are collected over newswire and web text from TAC Knowledge Base Population (TAC KBP) evaluations.

## 5 CONTEXT-GUIDED ANNOTATION SCHEME

**The Necessity of Human Annotation.** The main challenge to constructing an OOD RE dataset is how to guarantee the faithfulness of the evaluation results on a dataset when the model is trained on another dataset. There exist two questions: (1) Are their labels (relation types) the same? For example, given a model trained on dataset A with labels numbered from 1 to 10, it will be unfairly and unfaithfully evaluated by dataset B if the labels in dataset B are numbered from 11 to 20: The intersection between the two sets of labels is an empty set. Such a test becomes a zero-shot test instead of an OOD test. (2) How can we ensure that two given labels are the same? For example, there is a relation type in ACE05 named "ORG-Affiliation" (including Employment, Ownership, Founder, Student-Alum, Sports-Affiliation, Investor-Shareholder, and Membership) and another two relation types in DocRED named "employer of" and "member of", respectively. It is unfair to demand the models trained on DocRED to predict "ORG-Affiliation" accurately on ACE05 because these models haven't learned the knowledge about Founder and Investor-Shareholder. Furthermore, even if the two given relation types have the same name, they can represent different semantic situations. The issue stems from the semantic inequality between the given two labels (relation types). To address the issue, we must conduct human annotation.

**The challenges of Human Annotation.** We explore the existing human-annotated RE datasets and integrate the samples that explain the common relation to construct OODREB. The major challenge comes from the quadratic number of relation types with

regard to the total number of relation types in a dataset. Specifically, there are 263 kinds of relation types in the 9 candidate datasets. One of our goals is to identify and select all of the semantically identical relation types among them. For example, if we find clues that the relation label "residence" in DocRED is semantically identical to the relation label "Live-In" in CONLL04, then the models on DocRED can be used to predict the samples labeled with "Live-In" on CONLL04. The OOD performance of models can thereby be tested if the data in CONLL04 and DocRED are not independently and identically distributed. Note that a typical kind of wrong annotation easily occurs when two relation types possess a common semantic meaning. For example, the relation type *Physical* denotes that the subject entity is physically located in or physically near the object entity while the relation type *work location* also explains location information. Despite the common "location" meaning of the two types, they may describe different relations in certain situations, thus they are not semantically equal. To find these semantically identical relation types, we have to make $263 \times 263$ times of comparisons. To understand the exact meaning of each relation type and then distinguish whether two relation types from different datasets possess a common semantic meaning, we have to carefully read the context of each type of relation.

**Our Annotation Scheme.** To address the issues, we propose our context-guided annotation scheme. In the first step, we take all the relation types in DocRED to form the initial relation set of OODREB. We will compare the relation types in other datasets with the initial relation types. We adopt such a strategy in the first step for three reasons. First, the strategy significantly reduces the quadratic times of comparison. Second, DocRED contains the most relation types (96 in total) without repetition. Third, DocRED provides descriptions for each relation type, which mitigates the semantic ambiguity of relation types and thus makes it easier for annotators to make accurate annotations. In the second step, we attach additional 10 samples to each relation type as context to further help the annotators understand its exact semantic meaning. After reading the name of a relation type, its description, the attached samples, and the corresponding entity pair in the samples, annotators are able to capture the semantic meaning of the relation type, thus providing faithful annotation. In the third step, each annotator is required to distinguish whether each relation type in other datasets is semantically equal to that in the initial relation set. We preserve those relation types possessing the same semantic meanings to form OODREB and discard the other relation types. To guarantee the quality of annotation, we subject each annotation to one to two rounds of verification. If inconsistency occurs between the annotations from two annotators, the third annotator will check the semantic meaning again by retrieving another ten samples and reading twenty samples in total. Finally, 70,237 samples from 7 datasets with 96 relation types constitute OODREB.

# 6 EXPERIMENTS

## 6.1 Evaluated Models

We evaluate the following SOTA methods on long-text RE as they are able to tackle both intra-sentence RE and inter-sentence RE. We do not evaluate the sentence-level RE methods due to the lack of

their ability to tackle inter-sentence RE (e.g., DocRED in OODREB), which is a common scenario in real-world applications.

**SciBERT.** SciBERT [2] is a pre-trained language model that is pre-trained on scientific data based on BERT [9]. The model achieves significant performance gain in the scientific domain.

**ATLOP.** ATLOP [41] is the SOTA transformer-based method with adaptive thresholding and localized context pooling.

**DocuNet.** DocuNet [39] is the SOTA graph-based method using the captured local and global entity-level dependencies to facilitate document-level RE.

**KD.** KD [30] is pre-trained on distantly supervised data to tackle document-level RE and is the SOTA method on DocRED. The methods propose axis attention and the teacher-student network to distill the knowledge and improve model performance.

**EIDER.** EIDER [33] adopts human-annotated evidence sentences to supervise the model. The method first predicts evidence sentences from the given document and then guides the model to pay attention to evidence information.

## 6.2 Implementation Details

We apply AdamW algorithm [19] to optimize model parameters with $\beta_1 = 0.9, \beta_2 = 0.999$. All models are trained on 4 NVIDIA A10 GPUs. The batch size is set to 8.

## 6.3 Evaluation Metrics

We evaluate the robustness of models by the overall F1 score on the testing samples under three kinds of attacks, including mask entity attack (EM), randomly shuffled entity attack (ER), and unseen entity substitution attack (ES). We mask all entity names in EM. In ER, we randomly permute the entity names in the given document. In ES, we replace entity names in the given document with entities that have never occurred in the training set of DocRED. To evaluate the OOD generalization ability of models, we calculate the F1 score over our proposed OODREB. Following previous benchmarks [35], we also adopt Ign F1 to measure the F1 score that ignores the triples that appear in the training data.

## 6.4 Performance under *i.i.d.* Assumption

To exhibit the unique challenge in comparison with previous benchmarks, we first evaluate the performance of current SOTA RE methods on our proposed OODREB under the *i.i.d.* assumption, where we follow the previous benchmarking protocols for a fair comparison. Specifically, we divide OODREB into training, validation, and test data in an 8:1:1 ratio. In practice, we segment each of the 7 RE datasets in OODREB, with the randomly sampled 80% of each dataset forming the training data, and the remaining 20% divided equally to form the validation and test data. We evaluate the performance of models trained on the training data.

As shown in Table 2, even when processing OODREB under the *i.i.d.* assumption for training and testing, it presents a significant challenge to the existing SOTA methods. The existing SOTA methods on document-level RE, which are claimed to be able to tackle both intra- and inter-sentence RE, suffer from achieving a satisfying performance on the test data of OODREB. Specifically, all the methods achieve the performance of approximately 60% F1 score on DocRED, while their performances sharply drop by 24%

| Model | Dev | | Test | |
|---|---|---|---|---|
| | F1 | Ign F1 | F1 | Ign F1 |
| *DocRED* | | | | |
| ATLOP$_{BERT}$ | 61.09 | 59.22 | 61.30 | 59.31 |
| Eider$_{BERT}$ | 62.48 | 60.51 | 62.47 | 60.42 |
| KD$_{BERT}$ | 62.03 | 60.08 | 62.08 | 60.04 |
| DocuNet$_{BERT}$ | 61.83 | 59.86 | 61.86 | 59.93 |
| *OODREB* | | | | |
| ATLOP$_{BERT}$ | 64.87 | 63.02 | 36.50 | 34.60 |
| ATLOP$_{SciBERT}$ | 63.00 | 61.17 | 35.24 | 33.43 |
| Eider$_{BERT}$ | 64.08 | 62.26 | 35.39 | 33.49 |
| KD$_{BERT}$ | 47.10 | 44.85 | 25.67 | 23.84 |
| DocuNet$_{BERT}$ | 67.74 | 66.07 | 35.98 | 34.05 |

**Table 2: Results of the performance on DocRED and OODREB under *i.i.d.* assumption. The same trend occurs when evaluating RoBERTa-based models on the test set of OODREB. We omit their performance due to the limited space.**

to 36.2% F1 score on the test data of OODREB. We elaborate on our following findings and the corresponding underlying causes to rethink the developing protocol of relation extraction methods.

**Learning Difficulty of causality.** Even if we assume that the training data have comprised all the potential scenarios, including various lengths of context, diverse domains of content, and all types of relations, the existing SOTA RE methods still fail to learn the rationales or causal information from the given training data. The experimental results in Table 2 exhibit the high learning difficulty of the samples in OODREB. Compared with learning to perform RE from domain- and length-similar textual data (e.g., the training data from DocRED), it is more challenging for existing models to learn the rationales in RE from diverse distributions of the given training textual data. All the well-performed models (at least 60% F1 score on the test data) on DocRED suffer from effectively acquiring knowledge in the test data sampled from OODREB (no more than 36.5% F1 score). Therefore, merely expanding the human-annotated training data can be ineffective in developing the OOD generalization ability (the ability to acquire knowledge in the wild) of RE methods. We appeal to RE practitioners to put more attention on other aspects of developing a RE model, such as enhancing the causal learning ability of models through human knowledge infusion. Considering a simple linear regression model, for example, we can enhance its causal learning ability by setting the coefficients of all non-causal variables to zero.

**Model Selection Bias.** The significant gap between model performances on the validation data and test data in Table 2 indicates another crucial conclusion: The common design of the model selection stage is insufficient to select the well-performed models in real-world RE applications. Specifically, models can learn certain spurious patterns instead of human-aligned features. Current model selection methods select those models that successfully learn the predictive features in the validation data, skipping over the step of distinguishing whether the learned features are causal information or not. This limitation induces the model selection bias, which

leads to the failure of the selected models in test data. For example, DocuNet is considered well-performed by achieving 67.74% F1 score on the validation data, while its performance drops sharply by 33.69% on the test data. Therefore, rethinking the model selection methods is of great urgency.

To sum up, we appeal to future work to take both the causality learning ability of RE models and the development of model selection methods into consideration, which indicate the rethinking of the developing protocols of RE methods.

## 6.5 Robustness and Generalization Ability

We investigate the generalization ability and robustness of the models trained on the training set of DocRED in both *i.i.d.* and OOD settings. We consider the models in document-level RE because they can be applied to tackle the sentence-level RE task. They are expected to extract both inter- and intra-sentence relations, while models in sentence-level RE are demonstrated to struggle to extract inter-sentence relations [35]. The results of our evaluation are shown in Table 3. We observe a significant discrepancy in the generalization ability and robustness between i.i.d and OOD settings. More observations and analysis are as follows.

The performance of the models in the OOD setting can sharply drop by at most 40.55 F1-score compared to their performance in the *i.i.d.* setting. Due to the violation of the *i.i.d.* assumption and the lack of OOD generalization ability, the models struggle to predict the relations effectively when exposed to our proposed OODREB with different distributions (e.g., the average sentence length and the source of the data). The models show the strong sentence-level RE ability represented by Intra-F1, but the ability in the *i.i.d.* setting fails to generalize to OODREB. The ability can not facilitate improving their performance on OODREB even though it is composed of abundant sentence-level relations. Specifically, the sentence-level RE performance of ATLOP$_{BERT}$, ATLOP$_{RoBERTa}$, and Eider$_{RoBERTa}$ in the OOD setting strikingly drops by 46.72, 44.45, and 42.93 F1-score when compared to their performance in the *i.i.d.* setting. The results demonstrate that enhancing the generalization ability of the models in RE is of critical significance. Otherwise, the lack of this ability will not only hamper the models from acquiring knowledge in the wild but also impede their deployment in real-world scenarios.

We notice that KD$_{RoBERTa}$ achieves a better performance than other SOTA models on document-level RE. KD$_{RoBERTa}$ adopts additional distantly supervised training data compared with other models, which indicates that exposing models to more extra distantly supervised training data can seemingly be an effective method to improve their performance in both *i.i.d.* and OOD settings. We posit the underlying reason is that the augmented various linguistic expressions help models to capture the causal semantic features for relations. Despite the improved performance, KD$_{RoBERTa}$ still suffers from making accurate predictions.

The overall experimental results in Table 3 show that the models are vulnerable to all three kinds of attacks in both *i.i.d.* and OOD settings, indicating that the existing methods in document-level RE predict relations according to entity names in most situations. The vulnerability reveals that models fail to capture the rationales. That is to say, even though the OOD generalization performances of the

| Model | *i.i.d.* Setting | | | | | OOD Setting | | | |
|---|---|---|---|---|---|---|---|---|---|
| | EM-F1 | ES-F1 | ER-F1 | *i.i.d.*-F1 | Intra-F1 | EM-F1 | ES-F1 | ER-F1 | OOD-F1 |
| ATLOP$_{BERT}$ | 6.39 | 6.08 | 14.16 | 61.09 | 67.26 | 1.33 | 0.93 | 3.66 | 20.54 |
| ATLOP$_{RoBERTa}$ | 27.29 | 7.35 | 17.50 | 63.18 | 69.61 | 9.24 | 4.46 | 7.22 | 25.16 |
| DocuNet$_{RoBERTa}$ | 8.62 | 8.08 | 18.55 | 63.91 | - | 6.56 | 4.81 | 9.67 | 29.16 |
| Eider$_{RoBERTa}$ | 35.45 | 8.46 | 23.00 | 64.28 | 70.36 | 8.87 | 4.83 | 10.30 | 27.43 |
| KD$_{RoBERTa}$ | 29.74 | 7.57 | 20.35 | 67.12 | - | 29.31 | 14.01 | 24.62 | 43.57 |

**Table 3: Evaluation results of robustness and generalization ability in both *i.i.d.* and OOD settings. In the *i.i.d.* setting, we conduct experiments on the validation set of DocRED. In the OOD setting, we experiment on our proposed OODREB where the training samples are excluded. The overall performance on it can be represented by "OOD-F1". "*i.i.d.*-F1" and "Intra-F1" denote the overall and intra-sentence (sentence-level RE) performance on the validation set of DocRED, respectively.**

models are limited, a large part of them is fragile. Therefore, the development and study of the ability to generalize under various distributional shifts is of great urgency in RE.

The robustness of models in the OOD setting also exhibits a sharp drop compared to that in the *i.i.d.* setting. Specifically, the performances of models under EM, ES, and ER drop by at most 26.58, 5.15, and 12.7 F1-score, respectively. Compared with tackling OOD generalization, improving the robustness of models in the OOD setting is even more challenging.

### 6.6 Prompts and Performance of ChatGPT

Our prompt is written as follows:

Please extract the relational triplets from the following document according to the instructions. Note that the entity names have been enclosed in angle brackets.

Document: *<Skai TV> is a Greek free-to-air television network based in <Piraeus>. It is part of the <Skai Group>, one of the largest media groups in the country. It was relaunched in its present form on <the 1st of April 2006> in the <Athens> metropolitan area and gradually spread its coverage nationwide. Besides digital terrestrial transmission, it is available on the subscription-based encrypted services of <Nova and Cosmote TV>. <Skai TV> is also a member of <Digea>, a consortium of private television networks introducing digital terrestrial transmission in <Greece>. At launch, <Skai TV> opted for dubbing all foreign language content into Greek, instead of using subtitles. This is very uncommon in <Greece> for anything except documentaries (using voiceover dubbing) and children's programs (using lip-synced dubbing), so after intense criticism the station switched to using subtitles for almost all foreign shows.*

Instructions: the output format is [subject entity, relation, object entity] and there are 96 types of following candidate relations to be extracted: ['head of government', 'country', 'place of birth', 'place of death', 'father', 'mother', 'spouse', 'country of citizenship', 'continent', 'instance of', 'head of state', 'capital', 'official language', 'position held', 'child', 'author', 'member of sports team', 'director', 'screenwriter' ', 'educated at', 'composer', 'member of political party', 'employer', 'founded by', 'league', 'publisher', 'owned by', 'located in the administrative territorial entity', ' genre', 'operator', 'religion', 'contains administrative territorial entity', 'follows', 'followed by', 'headquarters location', 'cast member', 'producer', 'award received', 'creator', 'parent taxon', 'ethnic group', 'performer', 'manufacturer', 'developer', 'series', 'sister city', 'legislative body', 'basin

country', 'located in or next to body of water ', 'military branch', 'record label', 'production company', 'location', 'subclass of', 'subsidiary', 'part of', 'original language of work', 'platform', 'mouth of the watercourse', 'original network', 'member of', 'chairperson', 'country of origin', 'has part', 'residence', 'date of birth', 'date of death', 'inception', 'dissolved , abolished or demolished', 'publication date', 'start time', 'end time', 'point in time', 'conflict', 'characters', 'lyrics by', 'located on terrain feature', 'participant' , 'influenced by', 'location of formation', 'parent organization', 'notable work', 'separated from', 'narrative location', 'work location', 'applies to jurisdiction', 'product or material produced', 'unemployment rate', 'territory claimed by', 'participant of', 'replaces', 'replaced by', 'capital of', 'languages spoken, written or signed', 'present in work', 'sibling'].

The text written in *italics* represents the variable part of our template. This part varies according to the given documents.

**Performance of ChatGPT.** The example output of ChatGPT corresponding to the given prompt is as follows:

1. ["Skai TV", "is a member of", "Digea"],
2. ["Skai TV", "is part of"," the Skai Group"],
3. ["Skai TV", "is available on", "the subscription-based encrypted services of Nova and Cosmote TV"],
4. ["Skai TV", "opted for", "dubbing all foreign language content into Greek"],
5. ["Skai TV", "switched to", "using subtitles for almost all foreign shows"],

while the ground truth is as follows:

1. ["Piraeus", "country", "Greece"],
2. ["Skai Group", "country", "Greece"],
3. ["Athens", "country", "Greece"],
4. ["Skai TV"," headquarters location", "Piraeus"],
5. ["Skai TV"," owned by", "Skai Group"],
6. ["Skai TV"," headquarters location", "Athens"],
7. ["Skai TV", "country", "Greece"].

We can observe the unsatisfying performance of ChatGPT in this case. First, it does not understand our given instructions. Its extracted relation "opted for" and "switched to" does not appear in the candidate 96 types of relations. Second, it seems that ChatGPT only concerns about the relations of the first entity in the document. We require further prompts to guide it to pay attention to other entities and output more relational triplets. We leave the prompt

| Model | CoNLL OOD-F1 | FewRel OOD-F1 | KBP OOD-F1 | SciERC OOD-F1 | SemEval OOD-F1 | TACRED OOD-F1 | DocRED F1/Ign F1 |
|---|---|---|---|---|---|---|---|
| ATLOP$_{BERT}$ | 2.24 | 34.85 | 1.41 | 0.00 | 0.00 | 11.23 | 60.59 / 58.73 |
| ATLOP$_{RoBERTa}$ | 2.73 | 39.80 | 2.12 | 0.00 | 0.00 | 13.94 | 62.56 / 60.72 |
| ATLOP$_{SciBERT}$ | 2.00 | 22.41 | 0.18 | 0.00 | 0.00 | 5.80 | 57.92 / 56.20 |
| DocuNet$_{BERT}$ | 3.67 | 42.80 | 1.96 | 0.00 | 0.00 | 10.97 | 60.86 / 59.06 |
| DocuNet$_{RoBERTa}$ | 2.83 | 42.30 | 2.33 | 0.00 | 0.00 | 14.29 | 61.78 / 60.01 |
| Eider$_{BERT}$ | 1.91 | 31.89 | 1.42 | 0.00 | 0.00 | 10.29 | 61.26 / 59.46 |
| Eider$_{RoBERTa}$ | 2.71 | 43.50 | 3.09 | 0.00 | 0.00 | 14.94 | 62.93 / 61.04 |
| KD$_{RoBERTa}$ | 6.76 | 62.22 | 10.07 | 0.58 | 0.52 | 31.96 | 67.14 / 65.09 |

**Table 4: The performance of models trained on the training set of DocRED. All the experimental results are based on our implementation.**

engineering and potential unleashing of ChatGPT as future work. Third, ChatGPT fails to understand the semantic meaning of our prompts. While we claim that all the entities in the given document are enclosed in angle brackets and the output format is [subject entity, relation, object entity], ChatGPT still considers the phrase "dubbing all foreign language content into Greek" and "using subtitles for almost all foreign shows" as entities and extracts them as relational triplets.

The overall performance of ChatGPT on the validation set of DocRED is 0.921%, which indicates that both document-level relation extraction and real-world relation extraction remain challenging for large language models to acquire knowledge in the wild.

## 6.7 Performance on Datasets

In this section, we elaborate on the analysis of generalization ability on different subsets (datasets) in OODREB. Trained by the original training data in DocRED, models achieve various performances on the other 6 datasets. The 6 datasets are sampled from various domains and comprise significantly different lengths of context from DocRED. As shown in Table 4, we can observe the sharp performance drops of all models on the 6 datasets. One of the apparent underlying reasons is the significant shift in context lengths. Our further findings are as follows.

**Degree of Distributional Shifts Affects Model Performances.** Among the 6 datasets, model performances on FewRel present less severe drops by at least 18.06% F1 score. Since the textual data in FewRel and DocRED share a common background: They are sampled from Wikipedia and their candidate relation types present a significant overlap, the distributional shift from DocRED to FewRel is relatively minor. This explains the underlying reason why the model performances on FewRel are better than that on the other 5 datasets. Specifically, the textual data in SciERC are sampled from scientific papers, the writing style and domain of which make its distributional shift most significant compared with other datasets, thereby presenting the greatest degree of distributional shift and leading to the most severe performance drops of all models. None of the models trained by DocRED is able to make any correct predictions on SciERC, which reveals the severe lack of OOD generalization ability of current models. We have considerable work to do in the future to deploy them in real-world scenarios with various distributional shifts.

**Learning Causality is a Random Event.** All significant performance drops reveal the fact that current SOTA RE models fail to learn rationales or causal relationships between the semantic meanings explained by context and relations: Their learned spurious patterns are not stable, while causal relationships are stable across various scenarios [8]. Meanwhile, we observe that some models stably make a small number of right predictions, which indicates that they successfully learn a small portion of causal relationships. Hence, we consider learning causality as a random event, whose probability is affected by the initial values of model parameters. Specifically, when adopting the same model architecture, training data, objective function, and optimizer, ATLOP exhibits different OOD generalization performances due to the different initial values of their adopted pre-trained language models (PLMs). Similarly, we observe the significant performance improvement of KD$_{RoBERTa}$, which is pre-trained by the distantly supervised data to improve initial values. We posit a reason for the better performance of KD$_{RoBERTa}$: Correlations between the semantic meanings explained by context and relations are augmented in the distantly supervised data, which drives the model to learn such correlations. If there are no other correlations in each training sample, its learned correlations become causal relationships. Consequently, without any guidance or data augmentation to improve model parameters (e.g., PLMs) in advance, learning causality inherently becomes a random event due to the various correlations in the training data. Intuitively, considering the simple linear regression model (mentioned in section 6.4) with the coefficients of all non-causal variables set to zero (improving parameters by causality), we know that it can always learn causality if the training loss is minimized to zero.

## 6.8 Performance on Relation Types

We investigate model performance across all relation types and exhibit the results in Figure 2. Our findings are as follows.

**Difficulty in Learning Causality Exists and Varies across Relation Types.** As shown in Figure 2, we observe that some relation types are easier for a model to learn its predictive patterns while the other relation types are more difficult to be accurately predicted. Specifically, to demonstrate the difficulty in learning causality, we first present the correlations between performance and the number of training samples on the left of Figure 2, where the increasing training samples with the minimized training error do not lead

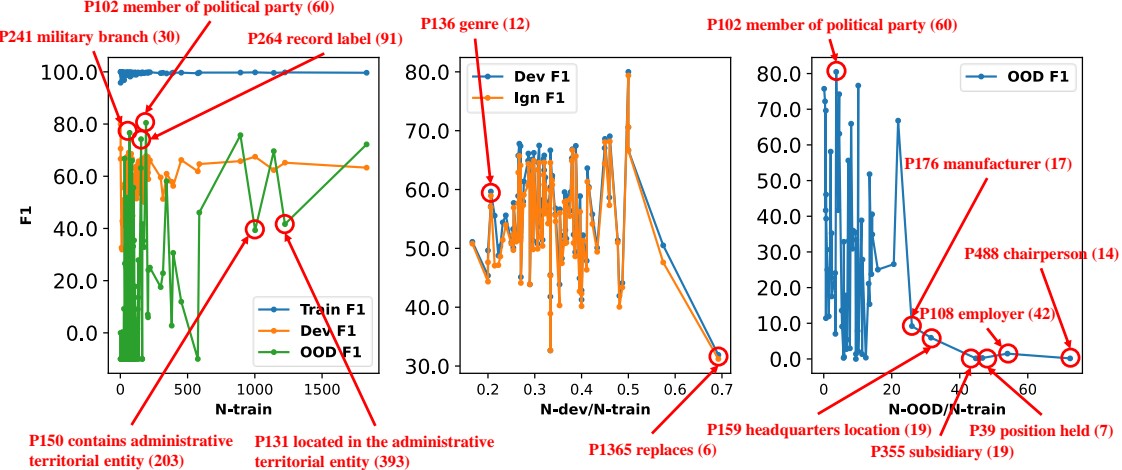

**Figure 2: Performance of ATLOP$_{\text{RoBERTa}}$ across all relation types. The numbers in parentheses indicate the number of various linguistic expressions referring to the corresponding relation type. The linguistic expressions are presented by human-annotated evidence words, which are provided by DocRED$_{\text{HWE}}$ [6] with the 700 samples from the original validation set of DocRED. The negative value of the F1 score indicates that the corresponding relation types do not occur in the training data.**

to better performance on relation types in either *i.i.d.* setting or OOD setting. The phenomenon is consistent with our finding in section 6.4: Since the training samples with various kinds of distributions are hard to learn, merely increasing their number becomes ineffective. Second, as shown again on the left of Figure 2, we observe poorer performance on some relation types with the same or a larger number of training samples compared with other relation types, which indicates that the difficulty in learning causality varies across relation types. Specifically, the semantic features that causally determine the P131 relation type "located in the administrative territorial entity" are ambiguous for the model to learn from the given more than 1,200 training samples, while the model successfully learns the predictive features of P241 relation type "military branch" through no more than 100 training samples. We elaborate on the underlying causes in the next paragraph.

**Influencing Factor of Difficulty in Learning Causality.** We take a step further toward diagnosing the underlying causes by investigating a crucial factor: the *diversity* of linguistic expressions of each relation type. We find clues that the more diverse the linguistic expressions are, the more difficult it is for models to capture the underlying common causal semantic features. Specifically, the P131 relation type has 393 kinds of linguistic expressions, which is intuitively difficult for learning models to capture the causal relationship between their common semantic features (e.g., "located in", "is in", "the position is", etc.) and the semantic meaning of the relation type "located in the administrative territorial entity". Similar to P131, P150 contains significantly more diverse linguistic expressions than other relation types, which renders the two relation types most difficult for models to understand despite their largest number of training samples. We refer readers to the appendix for more details.

**Real-World Scenarios are Challenging for RE.** The higher the ratio of OOD test samples to training samples, the closer the scenarios represented by the data align with real-world scenarios, as

RE practitioners have limited access to abundant human-annotated data training data and simultaneously require the RE models to be able to tackle the widest range of OOD data in real-world applications. As shown on the right of Figure 2, we observe that the model performs well on none of the relation types if their test data become increasingly reflective of real-world scenarios. Specifically, when the ratio of the number of OOD test samples to the number of training samples exceeds 20%, the model fails to both tackle the distributional shifts and make accurate predictions despite the fewer kinds of linguistic expressions of the current relation type. Their learned patterns in the training data are unstable to succeed in tackling real-world scenarios. This indicates again the importance of learning causal relationships (e.g., stable patterns). In the *i.i.d.* setting as shown on the middle of Figure 2, the ratio becomes higher than 50% due to the eased standards of finding $f_m$ without considering $\mathcal{D}$ mentioned in Section 3.

## 7 CONCLUSION

Acquiring knowledge in the wild is the ultimate goal of the RE task, which requires the out-of-distribution generalization ability of RE methods. Despite the recent emergence of LLMs with strong generalization ability, they still struggle in accurately extracting relations. Previous benchmarks neglect the evaluation of the OOD generalization ability of models, which renders the current SOTA RE methods far away from being deployed into real-world scenarios.

In this paper, we focus on selecting models that can be well-performed in real-world applications. To this end, we rethink the whole developing protocols of RE methods by proposing an out-of-distribution relation extraction benchmark (OODREB) and revealing new insights to inspire future work. Our experimental results also indicate that improving the OOD generalization ability of models is challenging and of great urgency.

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

## A PERFORMANCE ON RELATION TYPES

Details of model performance on all relation types are shown in Table 5.

Received 20 February 2007; revised 12 March 2009; accepted 5 June 2009

| Relation | N-total | N-train | N-dev | N-OOD | Total-F1 | Train-F1 | Dev-F1 | OOD-F1 | Ign F1 |
|---|---|---|---|---|---|---|---|---|---|
| head of government/P6 | 171 | 133 | 38 | 700 | 93.01 | 99.61 | 64.89 | 13.57 | 70.31 |
| country/P17 | 2417 | 1831 | 586 | 700 | 91.41 | 99.66 | 63.29 | 72.21 | 72.52 |
| place of birth/P19 | 588 | 453 | 135 | 669 | 92.01 | 99.68 | 66.23 | 11.93 | 76.79 |
| place of death/P20 | 220 | 170 | 50 | 392 | 91.89 | 99.63 | 65.1 | 35.25 | 77.19 |
| father/P22 | 205 | 164 | 41 | 700 | 91.04 | 99.55 | 51.06 | 41.7 | 70.39 |
| mother/P25 | 60 | 50 | 10 | 700 | 93.3 | 99.69 | 45.37 | 23.76 | 62.5 |
| spouse/P26 | 168 | 134 | 34 | 1527 | 91.09 | 99.71 | 53.29 | 27.86 | 69.77 |
| country of citizenship/P27 | 1526 | 1141 | 385 | 700 | 90.69 | 99.61 | 62.36 | 69.63 | 73.73 |
| instance of/P31 | 108 | 74 | 34 | 916 | 88.35 | 99.42 | 58.64 | 0.36 | 73.53 |
| position held/P39 | 21 | 15 | 6 | 700 | 87.83 | 100.0 | 42.86 | 0.28 | 68.14 |
| child/P40 | 222 | 177 | 45 | 1047 | 90.79 | 99.52 | 50.09 | 32.86 | 68.76 |
| director/P57 | 211 | 153 | 58 | 700 | 91.21 | 99.56 | 65.3 | 63.13 | 78.54 |
| screenwriter/P58 | 107 | 83 | 24 | 700 | 93.05 | 99.78 | 62.85 | 30.9 | 78.06 |
| educated at/P69 | 283 | 220 | 63 | 229 | 91.7 | 99.85 | 66.12 | 25.0 | 75.18 |
| composer/P86 | 65 | 44 | 21 | 700 | 84.45 | 99.71 | 51.36 | 25.03 | 72.0 |
| **member of political party/P102** | 242 | 191 | 51 | 700 | 93.55 | 99.75 | 67.79 | 80.48 | 76.55 |
| **employer/P108** | 156 | 126 | 30 | 6806 | 92.48 | 99.34 | 55.61 | 1.49 | 69.19 |
| founded by/P112 | 94 | 74 | 20 | 737 | 89.46 | 99.28 | 45.12 | 7.86 | 65.04 |
| league/P118 | 92 | 63 | 29 | 700 | 91.36 | 99.73 | 69.05 | 38.85 | 78.8 |
| publisher/P123 | 110 | 81 | 29 | 700 | 87.41 | 99.15 | 48.3 | 35.97 | 67.22 |
| owned by/P127 | 127 | 91 | 36 | 700 | 88.26 | 99.06 | 44.91 | 2.98 | 61.0 |
| **located in the administrative territorial entity/P131** | 1614 | 1224 | 390 | 700 | 91.94 | 99.78 | 65.2 | 41.66 | 74.26 |
| genre/P136 | 41 | 34 | 7 | 700 | 94.32 | 99.78 | 57.14 | 26.57 | 80.15 |
| operator/P137 | 71 | 52 | 19 | 700 | 88.34 | 99.74 | 53.11 | 15.36 | 70.12 |
| religion/P140 | 86 | 60 | 26 | 853 | 86.4 | 99.52 | 50.09 | 40.56 | 67.18 |
| **contains administrative territorial entity/P150** | 1312 | 1002 | 310 | 700 | 92.41 | 99.8 | 67.48 | 39.32 | 75.45 |
| follows/P155 | 160 | 117 | 43 | 700 | 88.43 | 98.88 | 51.65 | 0.57 | 69.15 |
| followed by/P156 | 158 | 120 | 38 | 700 | 89.73 | 99.17 | 51.43 | 0.28 | 68.97 |
| **headquarters location/P159** | 263 | 206 | 57 | 6496 | 92.67 | 99.78 | 61.36 | 5.95 | 73.91 |
| performer/P175 | 446 | 344 | 102 | 700 | 91.15 | 99.4 | 60.95 | 58.12 | 74.6 |
| **manufacturer/P176** | 36 | 27 | 9 | 700 | 88.41 | 96.79 | 45.5 | 9.21 | 59.62 |
| developer/P178 | 103 | 73 | 30 | 700 | 86.43 | 98.19 | 47.89 | 33.41 | 59.99 |
| located in or next to body of water/P206 | 144 | 109 | 35 | 700 | 90.27 | 99.78 | 58.2 | 17.8 | 69.46 |
| **military branch/P241** | 100 | 69 | 31 | 700 | 89.07 | 99.67 | 67.05 | 76.63 | 74.96 |
| **record label/P264** | 204 | 154 | 50 | 700 | 89.17 | 99.37 | 57.72 | 74.22 | 72.73 |
| location/P276 | 185 | 130 | 55 | 700 | 88.3 | 99.68 | 55.8 | 9.09 | 70.05 |
| subsidiary/P355 | 69 | 51 | 18 | 2281 | 87.02 | 99.08 | 43.87 | 0.17 | 55.12 |
| part of/P361 | 501 | 382 | 119 | 2558 | 89.87 | 99.45 | 57.89 | 2.76 | 72.71 |
| original language of work/P364 | 43 | 32 | 11 | 700 | 88.26 | 99.4 | 55.81 | 66.79 | 74.79 |
| platform/P400 | 66 | 52 | 14 | 700 | 90.07 | 98.77 | 51.85 | 51.8 | 64.14 |
| mouth of the watercourse/P403 | 68 | 49 | 19 | 700 | 90.72 | 99.37 | 67.43 | 34.9 | 75.83 |
| original network/P449 | 117 | 97 | 20 | 700 | 93.69 | 99.7 | 59.66 | 55.62 | 78.16 |
| member of/P463 | 263 | 208 | 55 | 700 | 91.56 | 99.73 | 58.91 | 24.09 | 73.55 |
| **chairperson/P488** | 64 | 49 | 15 | 3550 | 91.33 | 99.74 | 50.78 | 0.16 | 71.43 |
| country of origin/P495 | 413 | 300 | 113 | 700 | 89.28 | 99.51 | 59.57 | 17.52 | 73.77 |
| has part/P527 | 411 | 317 | 94 | 700 | 89.96 | 99.63 | 51.3 | 22.86 | 68.3 |
| residence/P551 | 30 | 25 | 5 | 5900 | 89.56 | 99.15 | 49.65 | 0 | 65.61 |
| **date of birth/P569** | 1179 | 893 | 286 | 103 | 91.73 | 99.73 | 65.8 | 75.74 | 76.39 |
| date of death/P570 | 767 | 587 | 180 | 394 | 91.79 | 99.72 | 64.67 | 46.09 | 74.88 |
| inception/P571 | 522 | 393 | 129 | 719 | 89.85 | 99.71 | 56.29 | 30.64 | 69.95 |
| dissolved, abolished or demolished/P576 | 77 | 52 | 25 | 33 | 83.32 | 99.74 | 41.94 | 11.43 | 61.1 |
| characters/P674 | 87 | 62 | 25 | 700 | 89.29 | 99.28 | 52.24 | 1.33 | 70.1 |
| located on terrain feature/P706 | 103 | 74 | 29 | 700 | 87.51 | 99.65 | 54.5 | 1.38 | 65.92 |
| participant/P710 | 117 | 95 | 22 | 700 | 91.25 | 99.48 | 54.41 | 33.13 | 63.43 |
| location of formation/P740 | 65 | 53 | 12 | 700 | 92.01 | 99.47 | 50.37 | 21.12 | 66.56 |
| parent organization/P749 | 74 | 47 | 27 | 444 | 84.78 | 99.36 | 50.51 | 0 | 63.47 |
| notable work/P800 | 134 | 102 | 32 | 700 | 90.18 | 99.0 | 54.76 | 5.39 | 71.23 |
| separated from/P807 | 3 | 2 | 1 | 974 | 90.91 | 100.0 | 70.59 | -100 | 77.52 |
| work location/P937 | 88 | 69 | 19 | 700 | 93.32 | 99.71 | 57.96 | 2.11 | 74.06 |
| applies to jurisdiction/P1001 | 259 | 204 | 55 | 700 | 93.08 | 99.66 | 67.38 | 7.04 | 75.3 |
| participant of/P1344 | 116 | 87 | 29 | 700 | 91.9 | 99.64 | 66.67 | 65.94 | 76.79 |
| sibling/P3373 | 128 | 102 | 26 | 950 | 91.4 | 99.29 | 57.72 | 35.46 | 69.15 |

**Table 5: The performance of ATLOP$_{\text{BERT}}$ on predicting various relation types. N-train and N-dev denote the number of training samples and validation samples from DocRED, respectively. N-OOD indicates the number of non-DocRED samples.**