# OpenReview forum: "OODREB: Benchmarking State-of-the-Art methods for Out-Of-Distribution Generalization on Relation Extraction"
_ACM.org/TheWebConf/2024/Conference — TheWebConf24_

### Official Review · Reviewer_FPxr · 2023-11-23

**Novelty:** 5
**Technical Quality:** 4

**Review:**

The paper proposes a benchmark dataset, built using different state-of-the-art (SOTA) datasets, to evaluate the performance of SOTA approaches in out-of-distribution (OOD) settings. The paper presents an interpretation of the OOD problem and the construction of the OODREB dataset. Several SOTA approaches are evaluated using the proposed datasets to discuss their generalization capability, especially in OOD settings.

My overall opinion is that the work is promising, but it requires a substantial amount of conceptual and technical work of polishing. As a consequence, I am inclined to recommend not to accept the paper in its current shape.


PROS

The paper addresses an important problem in information extraction, which has a particular impact on deploying solutions that can work in the wild (relevant especially when extracting information from multiple web sources).

The idea of building a dataset to benchmark RE approaches in OOD is important and can have a huge impact on future work on RE

The number and choice of the considered methods are reasonable and preliminary experiments with GPT somehow show that the problem is still relevant and not solved (although a citation to a recent ACL paper on this topic [1] is missing).

Overall, the work is promising and worth being pursued



CONS

If we consider the call for papers (https://www2024.thewebconf.org/calls/research-tracks/ ) the paper does not comply with requirements, because it fails to discuss the relevance of the proposed approach to the Web context. However, in this particular case, I will not consider this as a criterion for rejection because relation extraction, and, in particular, dealing with OOD, is used so frequently to extract data from web sources. However, the authors should consider the guidelines and discuss this relevance in the introduction.

Some aspects are not explained clearly enough, e.g., the i.i.d. evaluation settings and OOD evaluation settings. In general, the paper discusses extensively (and with several repetitions or unnecessary observations) some specific topics, but fails to explain the key aspects that are needed to fully understand the paper. E.g.: no overview of the objectives of the experimental evaluation is given, no justification for critical choices like the attack strategies (why are they reasonable?), the explanation of the differences between experiments in Section 6.7 and other experiments is unclear, the requirements that models developed to consider a given context length should perform similarly on very different context lengths is not motivated enough.

Some theoretical aspects are not well-elaborated; for example, the reference to “causality” is quite debatable here; the choice of attack strategies is questionable from a linguistic point of view; the imputation of performance drop in i.i.d. evaluation settings is not completely well motivated (no discussion about other potential reasons is provided).



* Detailed comments on clarifications and other technical concerns

While I quite appreciate the formalization proposed in the paper, I also have some concerns. In the case in which a new specific domain is considered, the distribution changes so much that I am not sure we can assume humans apply the same decision rule model. As a matter of fact, the decision rule model by non-experts on a specialized medical or legal domain may differ from the one by expert users. I have the impression that the OOD problem addressed in this paper refers more to the linguistic style in different sources than to domain specificity. This issue and these definitions should be better scoped and discussed.

The paper imputes the lack of generalization in OOD settings to a vague factor of causality, which is not properly defined or explained in the paper, and is not grounded in a proper linguistic analysis. There are many different models of causality, which model of causality is referred to here? Some sort of linguistic analysis comes at the very end of the paper and is limited to the number of linguistic expressions used. This is evident, in my opinion, from the sentence “we observe poorer [...]” in Section 6.8; it is quite obvious that P131 is semantically similar to several relations and can appear with confounding linguistic patterns, while P241 has a quite specific context (military). This is acknowledged in the following paragraph, but it is not clear in which sense these factors are related to some form of causality and is also limited to discussing the number of linguistic expressions.

The claim that document-level and sentence-level RE can be mixed in the evaluation should be explained better (possibly, with some examples).

I feel a lack of theoretical framework in the paper. For example, the problem of having different relations is the problem of establishing mappings between relations with some agreed semantics (two relations may be equivalent, one more specific than another one, etc).  Observe that in principle, a model trained to predict a relation P can also predict a relation Q, if P is more specific than Q. I understand that considering this would complicate things further, and I think it is ok to focus on equivalence relations, but, again relations have some mathematical interpretation that is relevant to the problem you are addressing here and not fully considered in the paper. Another issue related to this one is that you do not specify well which is the cardinality expected for the mapping and why this choice is made; since you consider the larger set of DocRED relations, I imagine that models trained on a different dataset can only predict a subset of the 96 relations. Thus an important information that is not reported is how many relations in a source dataset X are mapped (as equivalent) to some relation in DocRED.

The choice of applying random shuffling or replacement should be further motivated. First, it seems based on the assumption that models should not leverage world knowledge at all in the evaluation, which is a strong assumption not explicitly formulated when you discuss OOD. Second, even if you want to push models to use as little world knowledge as possible, I am concerned that naming patterns and verb usage may clash and confuse the models. For example, I expect that a sentence like “Apple Inc work for Lake Michigan” would confuse even human annotators, because of “inc” and “lake” and the expectations that the verb work related a person with a company. A better strategy in my opinion would be to constrain shuffling and replacements to comply with the original entity types. In this way, you could reduce the effect of world knowledge, without clashing with linguistic patterns that a model is expected to learn and do not strictly depend on specific data distributions.

* Other comments

Page 3 “We give the definition” : but this is not an operational definition. Your claim here is that your benchmark approximate this definition (see also concerns about this definition, expressed above).

The paper mentions causality in a few places; however, the role of causality in this context is never defined or explained; what are the cause-effect dependencies relevant in this context?

[1] Wadhwa, Somin, Silvio Amir, and Byron C. Wallace. "Revisiting relation extraction in the era of large language models." arXiv preprint arXiv:2305.05003 (2023). [https://aclanthology.org/2023.acl-long.868.pdf]

**Questions:**

Is there a complete mapping between the original datasets and OODREB, i.e., all relations in the original datasets are represented by some relation in OODREB?

In the i.i.d. evaluation on OODREB, are algorithms trained eventually on the same amount of training samples they were trained on when evaluated in the original papers? Could be the case that training data are just too limited to reach the performance reported in the original papers?

In the i.i.d. evaluation on OODREB are algorithms trained on each sub-partition and tested on each sub-partition or are they trained on the whole OODREB data?

There are many different models of causality, which model of causality is referred to here?

**Ethics Review Description:**

No ethics alert for this paper

**Reviewer Confidence:**

3: The reviewer is confident but not certain that the evaluation is correct

**Scope:**

3: The work is somewhat relevant to the Web and to the track, and is of narrow interest to a sub-community

---

### Official Review · Reviewer_zo18 · 2023-11-23

**Novelty:** 5
**Technical Quality:** 6

**Review:**

# Review of "OODREB: Benchmarking State-of-the-Art methods for Out-of-Distribution Generalization on Relation Extraction"

- This paper addresses issues in evaluating relation extraction models, specifically in avoiding the trap of non-realistic evaluation benchmarks. It emphasizes the need for benchmarks that reflect out-of-distribution (OOD) data, moving away from in-distribution datasets. This is because benchmarks based on independent and identically distributed (IID) data do not realistically reflect long-tailed distributions in real-world data.
- The paper proposes the development of a method to generate OOD benchmarks for evaluating models, with minimal human adaptation efforts. It integrates various datasets to create comprehensive OOD benchmarks.
- The paper finds that models underperform in OOD settings compared to typical IID settings, highlighting the inadequacy of current state-of-the-art methods for effective relation extraction. The paper has a specific focus on LM-based extraction, and critiques immersion-extraction models, suggesting a significant gap in their performance as revealed by the study.
- I found the paper to be well-written, approachable, and innovative. I believe it can have a positive impact on the field by addressing the need for better evaluation methods. I further believe the paper is suitable for the conference audience, as it balances technical depth with accessibility. It highlights the urgent need for improved evaluation benchmarks in relation extraction, and provides evidence that current methods need substantial improvement to handle real-world, diverse data distributions.

## Pros
- The paper introduces a novel method for generating out-of-distribution benchmarks, addressing a significant gap in relation extraction evaluation.
- It emphasizes realistic data distributions, moving away from the limited scope of IID (independent and identically distributed) datasets, making the research more applicable to real-world scenarios.
- The integration of various datasets for creating comprehensive benchmarks ensures a more thorough evaluation of models.
- The paper provides valuable insights into the performance limitations of current state-of-the-art models, particularly in OOD settings.
- It highlights the need for models to learn causality for effective relation extraction, a crucial aspect often overlooked in current methods.
- The paper is well-written and approachable, making it suitable for a wide range of audiences, including those who might not be deeply technical.

## Cons
- The method for generating OOD benchmarks, while innovative, might be complex to implement in practice, requiring integration of multiple datasets.
-  The focus on language model-based extraction models limits the scope of the paper, not addressing other types of models that could be used for relation extraction.
- The approachable nature of the paper, while a strength, might risk oversimplifying the complexities involved in relation extraction and benchmark generation.

**Questions:**

- One of the challenges in applying LLMs is their tendency to overproduce on extraction tasks, i.e. they generate triples that are not in the benchmark. Half of the problem there is what is termed hallucination; i.e. triples that are incorrect and not supported by the evidence. The other half of the problem are triples that are factually correct, but due to their absence in the benchmark are treated as incorrect. Using an approach where the benchmark is the gold standard, the latter case is one where the LLM's extraction will be considered an error when in fact it is not. How would your approach mitigate the latter problem?

**Reviewer Confidence:**

3: The reviewer is confident but not certain that the evaluation is correct

**Scope:**

3: The work is somewhat relevant to the Web and to the track, and is of narrow interest to a sub-community

---

### Official Review · Reviewer_R4m5 · 2023-11-24

**Novelty:** 4
**Technical Quality:** 5

**Review:**

The approach proposes a new benchmark for Relation Extraction focused on improving the Out-of-distribution generalization by combining existing datasets. The authors also use manual human evaluation to avoid problems. This can help researchers/developers in the future to test new approaches.

**Questions:**

None, the paper is clear and the approach is well explained.

**Reviewer Confidence:**

3: The reviewer is confident but not certain that the evaluation is correct

**Scope:**

3: The work is somewhat relevant to the Web and to the track, and is of narrow interest to a sub-community

---

### Official Review · Reviewer_8KHU · 2023-11-25

**Novelty:** 5
**Technical Quality:** 5

**Review:**

**Strengths:**
S1. The paper presents a new dataset that is generated by manually combining seven existing relation extraction (RE) datasets. Then, the authors conduct multiple experiments on existing RE methods to test their abilities on generalization.

S2. The discovery seems useful as it shows some underlying factors affecting the poor results of existing approaches.

S3. The proposed dataset is useful for the community.

**Weaknesses:**
W1. I find that the notion of out-of-distribution is not well-defined in the paper. It seems that the authors imply that examples coming from the seven datasets are from different distributions, and some may be more dissimilar than others. However, I don’t know if it is always the case for all relations, such as location or place of birth.

W2. I cannot find a link to check the dataset in the paper. I am interested to see how some relations such as located in the administrative area (P131) and location (P276) are mapped to relations in other datasets (Wikidata instructs that we use P131 for administrative area and P276 for non-administrative area – thus it’s hard to find a semantically identical relation).

W3. I think the paper would be better if they can provide some motivating examples about causality and how learning causality can help RE methods improve on OOD.

**Questions:**

Please see Weakness 2.

**Reviewer Confidence:**

2: The reviewer is willing to defend the evaluation, but it is likely that the reviewer did not understand parts of the paper

**Scope:**

3: The work is somewhat relevant to the Web and to the track, and is of narrow interest to a sub-community

---

### Official Review · Reviewer_evn4 · 2023-11-27

**Novelty:** 4
**Technical Quality:** 4

**Review:**

This paper presents an interesting study on out-of-distribution (OOD) relation extraction. In order to evaluate existing SOTA approaches on OOD relation extraction, the authors first built an OOD dataset, called OODREB, by leveraging several current benchmark datasets. During the process, the authors used the relations in one dataset as the base and try to find matching relations from the other datasets (to make sure the relations in the new dataset are representing the same semantics). Then, the authors conducted the evaluation on both the original  dataset and OODREB. The authors also carried out some generalization study and experimented with ChatGPT. Generally speaking, it is an interesting read and it also points out an important problem.

Pros:
1) An interesting and important study for out-of-distribution relation extraction.

2) A carefully created dataset to evaluate OOD relation extraction of different models.

**Questions:**

One main question is on the novelty. I understand that the authors are studying the OOD setting for relation extraction and constructed a new dataset. That being said, the novelty is still rather limited. I will leave this to the chairs to decide.

A few questions after reading the paper:

1) It might be good to indicate in Table 1 whether the numbers are number of sentences or documents. After reading the texts, it seems most of them are sentence-level samples?

2) In Section 6.2, I assume these values were determined using the Dev set?

3) For the second half of the first paragraph of Section 6.4, I think it would be sufficient to say "the splits were stratified according to the subsets."

4) For the human annotation, did the authors measure the agreement level between different annotators?

5) I have one confusion about the big difference between Dev and Test for OODREB: Since the train/dev/test were stratified according to the individual subsets, I would have expected the different methods to perform relatively comparable between Dev and Test for OODREB. Even the authors mentioned this was probably due to the Dev set was not sufficiently representative, it was still a bit surprising. Did the authors shuffle the data multiple times and examine this observation on different shuffles? E.g., 10-fold cross validation or simply different shuffles.

**Reviewer Confidence:**

3: The reviewer is confident but not certain that the evaluation is correct

**Scope:**

4: The work is relevant to the Web and to the track, and is of broad interest to the community

---

### Decision · Program_Chairs · 2024-01-22

**Decision:**

Accept

**Comment:**

The paper makes a valuable contribution in terms of benchmarking, particularly in the realm of out-of-distribution (OOD) relation extraction, releasing a dataset created by leveraging various current benchmark datasets. The creation of this benchmark provides a valuable resource for the community, shedding light on the limitations of current state-of-the-art models, especially in OOD settings.

 The authors addressed some of the reviewers' concerns about the novelty of the paper, the well-defined notion of out-of-distribution, accessibility to the dataset, the absence of motivating examples and theoretical aspects on causality, the depth of the analysis, etc, . The authors provide constructive satisfactory answers for refining the paper's clarity and theoretical underpinnings, while emphasizing their pioneering role in addressing the OOD problems of real-world RE models, being that also the reason they start with conducting a human-annotated accurate benchmark .

 To the authors, please do make sure the extensive answers to the reviewer's comments are properly included in the camera ready.